# PLC-Based Arrayed Waveguide Grating Design for Fiber Bragg Grating Interrogation System

**DOI:** 10.3390/nano12172938

**Published:** 2022-08-25

**Authors:** Ke Li, Pei Yuan, Lidan Lu, Mingli Dong, Lianqing Zhu

**Affiliations:** 1Key Laboratory of the Ministry of Education for Optoelectronic Measurement Technology and Instrument, Beijing Information Science & Technology University, Beijing 100192, China; 2Beijing Laboratory of Optical Fiber Sensing and System, Beijing Information Science & Technology University, Beijing 100016, China; 3Beijing Key Laboratory of Optoelectronic Measurement Technology, Beijing Information Science & Technology University, Beijing 100192, China

**Keywords:** fiber Bragg grating interrogator, arrayed waveguide grating, planar lightwave circuit, silicon photonics

## Abstract

A fiber Bragg grating (FBG) interrogator is a scientific instrument that converts the wavelength change of FBG sensors into readable electrical signals. To achieve miniaturization and integration of FBG interrogator, we designed and fabricated a 36-channel array waveguide grating (AWG) on silica-based planar lightwave circuits (PLC) as a key device in a built FBG interrogation system. It is used to achieve continuous demodulation in C-band, while maintaining high resolution. This AWG has a 1.6 nm channel spacing, 3-dB bandwidth of 1.76 nm, non-adjacent channel crosstalk of −29.76 dB, and insertion loss of 3.46 dB. The dynamic range of the FBG interrogation system we built was tested to be 1522.4–1578.4 nm, with an interrogation resolution of 1 pm and accuracy of less than 1 pm in the dynamic range of 1523.16–1523.2 nm. The test results show that the FBG interrogation technology, based on AWG, can realize FBG wavelengths accurately demodulated, which has high application value in aerospace, deep sea exploration, and environmental monitoring, as well as other fields.

## 1. Introduction

Fiber Bragg grating (FBG) [1,2,3,4,5,6,7,8] has become one of the hottest areas of research in the field of optical sensing in recent years, due to its widespread use in aerospace, deep-sea exploration, environmental monitoring, and other industries as an “electronic eye” to monitor temperature, strain, concentration, and other changes in the state of the external environment. To demodulate the wavelength of FBG and obtain the wavelength encoding temperature or pressure variation of FBG, researchers [9] cleverly combined photonic integrated circuit (PIC) [10,11,12,13,14] technology and FBG interrogation technology in 2004 to create an FBG interrogation system based on PIC technology. A PIC-based interrogation system provides exceptional benefits in the miniaturization and integration of FBG interrogators, as well as a compact structure and low power consumption, while maintaining high accuracy and resolution. It can be used in aerospace applications, such as microgravity/orbital position monitoring, radiation measurement, and high-frequency vibration testing. Array waveguide grating (AWG) is the core filtering device of the FBG interrogation system, based on PIC technology, and demodulation using different interrogation methods, including edge filtering method [15], center-of-mass detection method, relative intensity method [16], etc. Among them, the relative intensity method uses two signals to demodulate an FBG wavelength and is theoretically independent of light source output optical power jitter, with relatively high accuracy. At the same time, a theoretically high demodulation rate is possible, due to the absence of mechanical moving elements in this interrogation technique.

AWG can be fabricated on the following material systems [17,18]: silica-based planar lightwave circuits (PLC), silicon on insulator (SOI), indium phosphide (InP), and polymers. In 2014, Technobistft-fos [19] implemented a specially designed AWG on InP material with a dynamic range of 4000 micro-strain (4.8 nm) and wavelength resolution of 5 pm. InP-based waveguide devices, however, exhibit some polarization dependence and necessitates the employment of specialized external polarization processing, adding to the system’s complexity. In 2018, Tianjin Polytechnic University [20] designed an ultra-small AWG on an SOI substrate with a good transmission spectrum and high polarization sensitivity, with high accuracy in the range of 10 to 50 °C. However, SOI-based waveguide devices are single-polarized, coupling loss with optical fiber is large, and the small process tolerances make it impossible to obtain satisfactory performance. Current research has made efforts to improve wavelength resolution and increase dynamic range. However, they have a common problem when the center wavelength of FBG is close to the center wavelength of AWG. Only one channel can detect significant output optical power, there is an interrogation blind area, the demodulation dynamic range is constrained, and continuous C-band demodulation cannot be achieved. To solve these issues, the silica-based PLC platform is extremely attractive for passive components such as AWG, considering its low waveguide loss, low coupling loss with optical fiber, low polarization correlation, low cost, and mature preparation process. In this paper, we chose the suitable adjacent output waveguide spacing and suitable length of the free-propagation region (FPR) for the AWG to increase the bandwidth of each output channel of the AWG. As a result, the AWG spectrum had a significant amount of channel overlap to enable continuous FBG sensor demodulation in the C-band. We designed and fabricated a high-performance 36-channel AWG, with a channel spacing of 1.6 nm on PLC, thus enabling continuous C-band demodulation, while maintaining high resolution.

## 2. Design and Simulation

A wide-spectrum light source, circulator, FBG sensors, AWG, photodetector (PD) array, and readout circuit make up the AWG-based FBG interrogation system [21]. The connection is shown in Figure 1. The light signal from the broad-spectrum light source arrives at the FBG through the circulator, the narrow-band Gaussian light of a certain wavelength is reflected by the FBG, and the Gaussian beam enters the AWG through the circulator. The wavelength division multiplexing (WDM) function of AWG enables filtering. A PD is attached after each AWG output channel, from which the optical signal is transformed into an electrical signal. A readout circuit is utilized to process the change in optical power of each AWG output channel. Finally, the interrogation function finds the wavelength change of the optical signal reflected from the FBG and completes the demodulation.

The interrogation function is:(1)$$\ln\left(\frac{P_{n+1}}{P_{n}}\right)=\frac{8(\ln2)\Delta \lambda }{\Delta \lambda_{FBG}^{2}+\Delta \lambda_{n}^{2}}\lambda_{FBG}-\frac{4(\ln2)(\lambda_{n+1}^{2}+\lambda_{n}^{2})}{\Delta \lambda_{FBG}^{2}+\Delta \lambda_{n}^{2}}$$

The logarithm of the ratio of the output optical power of adjacent channels of AWG is linearly related to the central wavelength of the FBG (*λ_FB_**_G_*), where ∆*λ* is the offset of the central wavelength; *λ**_n_*, *λ**_n_*_+1_, and *λ_FB_**_G_* are the peak wavelengths of AWG’s channels *n*, *n* + 1 and the FBG spectra; and ∆*λ**_n_* and ∆*λ_FB_**_G_* are the half-peak bandwidths of the AWG and the FBG spectra. *P_n_* and *P_n_*_+1_ are the output power of AWG’s channels *n* and *n* + 1, which are the integral of the product of the emission spectrum of broad-spectrum light source’s emission spectrum, FBG’s reflection spectrum, and transmission spectrum of AWG’s adjacent channels. When demodulating different FBGs with different bandwidths(↑), the dynamic range(↑) and demodulation resolution(↓) of each FBG are also different.

*λ_FB_**_G_* will drift with changes in external temperature or pressure. When *λ_FB_**_G_* drifts between channels *n* + 1 and *n* + 2, the interrogation function is:(2)$$\ln\left(\frac{P_{n+2}}{P_{n}{}_{+1}}\right)=\frac{8(\ln2)\Delta \lambda }{\Delta \lambda_{FBG}^{2}+\Delta \lambda_{n+1}^{2}}\lambda_{FBG}-\frac{4(\ln2)(\lambda_{n+2}^{2}+\lambda_{n+1}^{2})}{\Delta \lambda_{FBG}^{2}+\Delta \lambda_{n+1}^{2}}$$

Therefore, multiple AWG’s channels can be used to demodulate an FBG.

The basic structure schematic of this AWG is shown in Figure 2. Input/output waveguides, input/output slab waveguide FPR, and arrayed waveguides are the five sections of the AWG. The slab waveguide FPR adopts a Rowland circle structure, and the multiplexed optical waves containing multiple wavelengths are coupled into the input waveguide, diffracted in the input slab waveguide, and coupled into the arrayed waveguides. The arrayed waveguides end face is uniformly arranged on the grating circumference, so the diffracted light arrives at the arrayed waveguides end face with the same phase; then, after the length difference of ∆*L* array waveguides transmission, different wavelengths of light have different phase differences, and then the output slab waveguide focuses the light at different output waveguide positions and outputs it, in order to complete the demultiplexing function.

The design of AWG follows the following equation [22]:(3)$$n_{c}\cdot \Delta L=m\cdot \lambda_{0}$$
(4)$$FSR=\frac{\lambda_{0}\cdot n_{c}}{m\cdot n_{g}}$$
(5)$$\Delta \lambda =\frac{\Delta x_{o}\cdot n_{s}\cdot d\cdot n_{c}}{R\cdot m\cdot n_{g}}$$
(6)$$n_{g}=n_{c}-\lambda \cdot \frac{dn_{c}}{d\lambda }$$
(7)FSR≥N⋅Δλ
where *n_c_* is the effective refractive index, ∆*L* is the difference between adjacent waveguide lengths, *m* is the diffraction orders, *λ*_0_ is the central wavelength of the AWG, FSR is the free spectrum range of the AWG, *n*_g_ is the refractive index of the AWG, ∆*λ* is the adjacent channel spacing, ∆*x*_o_ is the adjacent output waveguide spacing, *n*_s_ is the slab waveguide refractive index, *d* is the adjacent waveguide spacing, *R* is the Rowland circle diameter, and N is the number of output channels.

We discuss how the channel bandwidth is impacted by the distance between the adjacent output waveguides and length of the FPR. We completed the relevant simulations using the beam propagation method (BPM) solver in the WDM Phasar (v.2.0; Optiwave Systems Inc. Ottawa, ON, Canada) software, and the basic values were set according to Table 1. The adjacent output waveguide spacings were set to 12, 14, and 16 µm, respectively, and the light intensity data output from channel 19 was processed for the three cases to obtain Figure 3. Figure 3 depicts how the channel bandwidth changes, depending on the distance between adjacent output waveguides. While other parameters are left the same, the channel bandwidth increases as the distance between adjacent output waveguide decrease. Conversely, a reduction in the distance between adjacent output waveguides raises the coupling rate between waveguides, which has an immediate impact on the performance of the AWG and leads to an increase. The distance between adjacent output waveguide values will, thus, have a significant influence on the channel bandwidth.

The FPR lengths were set to 5300, 5500, and 5700 µm, and the light intensity data outputs from channel 19 in the three cases were processed to obtain Figure 4. The impacts of various FPR lengths on the channel bandwidth are depicted in Figure 4. With other parameters staying the same, the channel bandwidth rose as the FPR length increased or decreased. At the same time, the AWG’s channel loss increased. As a result, the channel bandwidth was impacted by the change in the FPR length.

Taking the aforementioned elements into account, the final design parameters of AWG are indicated in Table 1.

According to the AWG design parameters, the AWG spectra, obtained by simulation using the BPM solver, are shown in Figure 5. According to the simulation results, the AWG channel’s half-peak bandwidth was around 1.875 nm. The insertion loss of the middle output channel was the least and insertion loss of the two side output channels was the biggest, due to the presence of loss non-uniformity. The simulation findings, therefore, demonstrate the effect that the insertion loss changes with the channel, with the least insertion loss being approximately 1.1 dB; the maximum insertion loss was 3.8 dB. In the actual manufacturing process, a special process was used to lessen the loss’s non-uniformity.

We have simulated the coupling loss of optical fiber and waveguide using the finite-difference time-domain (FDTD) methodology in the Lumerical (2020 R2.3; Ansys Canada Ltd. Waterloo, ON, Canada) software. Figure 6a shows the modeling of optical fiber and waveguide coupling, and Figure 6b shows its optical field distribution. From the simulation results, the coupling loss of optical fiber and AWG was 0.1 dB, which can prove that the coupling loss of the optical fiber and AWG on SiO_2_ materials was low.

## 3. Experimental Measurement

### 3.1. Fabrication

The AWG was fabricated according to the parameters listed previously, and it was manufactured by Henan Shijia Photons Technology Co., Ltd. (Hebi, China). The chip is built on a PLC platform with a 6 µm top Si layer. Figure 7 depicts the fabrication process. The SiO_2_ lower cladding layer was first produced by wet oxygen oxidation at high temperatures. Plasma-enhanced chemical vapor deposition (PECVD) was used to epitaxially form the core layer, and GeO_2_ doping was necessary to achieve a refractive index of 0.75 percent throughout the deposition process. After that, the photolithographic pattern was transferred to the wafer using inductively coupled plasma (ICP) dry etching, and the waveguide core layer was etched to form a high-quality waveguide in the core region. Finally, PECVD was used to create a 6 µm thick SiO_2_ top cladding layer to protect the AWG. Figure 8 shows microscopic images of the fabricated AWG.

### 3.2. Measurement

To determine whether temperature will affect the measurement results, we have collected the spectra of AWG channel 1 at different temperatures and obtained the results shown in Figure 9. Figure 9a shows the transmission spectrum of channel 1 at different temperatures, and Figure 9b shows the relationship between the center wavelength of channel 1 and temperature. From the results, it can be seen that the center wavelength of the transmission spectrum of AWG red-shifted with the increase in temperature, and the temperature drift was 9.59 pm/°C, which affected the measurement results. Therefore, we conducted the experiments in an ultra-clean room with a constant temperature.

A test setup comprised an amplified spontaneous emission (ASE) wide-spectrum light source, six-axis adjustment platform, splitter, optical spectrum analyzer (OSA), optical power meter, and motion controller, and a PC was constructed to evaluate the performance of the produced AWG (see Figure 10). The ASE wide-spectrum light source was connected to the AWG’s input, and the AWG’s output was connected via a single-mode fiber to a splitter, which was then linked to OSA and an optical power meter. The spectrometer detects the transmission spectrum waveform of the AWG output channel, the optical power meter measures the optical power of the AWG output channel, and the input and output coupling positions are changed by the PC and motion controller.

Figure 11a displays the AWG transmission spectrum. Figure 11b shows the non-adjacent channel crosstalk between each channel of the AWG, while Table 2 displays the detailed performance analysis. The findings reveal that our constructed AWG performs reasonably well, and we achieved a large bandwidth transmission spectrum with a good Gaussian shape. The non-adjacent channel crosstalk was −29.76 dB, each channel was less than −25 dB, the insertion loss range was 3.46 dB, and the 3-dB bandwidth of AWG reached 1.76 nm.

To test the interrogation performance of this AWG, we built a discrete fiber grating interrogation system, based on the AWG, as shown in Figure 12, including the tunable laser, AWG, and optical power meter. The test was conducted using a single narrowband long light output from a tunable laser, instead of the reflected light from the FBG sensing. The light from the tunable laser transmitted to the AWG; the two adjacent output channels of the AWG were chosen to be connected to the two input ports of the power meter, and the output optical power values of the two adjacent channels of the AWG were recorded. Each device was connected to the other via optical fiber, and the central wavelength value of light emitted from the tunable laser and output power values of two adjacent channels of AWG detected by the power meter were recorded in real-time.

The interrogation findings of the experiment are displayed in Figure 13 and include the output optical power variation values for the AWG channels 1, 2, and 3, as well as channels 34, 35, and 36. Table 3 shows the central wavelengths of AWG channels 1, 2, 3, and 36. Due to the absence of vibration dampening measures on the operating table, some sharper edges appearred in channel 34 in Figure 13b, but this does not affect the conclusion. From the results, the dynamic range achievable by this test system was 1522.4–1578.4 nm. We also evaluated the continuity of interrogation. Figure 13a shows that AWG channels 1 and 2 have a continuous demodulation range of 1522.4–1524.0 nm, while AWG channels 2 and 3 have a continuous demodulation range of 1524.0–1525.6 nm. As a result, from 1522.4–1525.6 nm, AWG channels 1, 2, and 3 can be continuously demodulated. The FBG wavelength derived from the interrogation function was:(8)$$\lambda_{FBG}=\{\begin{array}{c} 1.8295\ln\frac{P_{2}}{P_{1}}-2786.7851\\ 1.7817\ln\frac{P_{3}}{P_{2}}-2716.7119 \end{array}$$

The results are in accordance with the designed channel spacing value of 1.6 nm. The continuous demodulation range will increase by 1.6 nm for each extra AWG output channel, so the interrogation system can achieve continuous demodulation in C-band. In practice, a 36-channel AWG with a channel spacing of 1.6 nm can demodulate m (1 ≤ *m* ≤ 18) FBGs using *n* (2 ≤ *n* ≤ 36) channels, and the dynamic range of a single FBG is 1.6 × (*n* − 1) nm.

We also tested the wavelength resolution of this system, with a wavelength scan range of 1523.16 to 1523.2 nm and a scan step of Δ*λ* = 0.001 nm. The photocurrent at each step was recorded by an optical power meter, and the calculated optical power ratio ln(*P*_2_/*P*_1_)—wavelength is shown in Figure 14. This measurement result shows the wavelength resolution of our AWG-based FBG interrogation was 1 pm. The resolution was 1 pm. The root means square (RMS) of the fitted line and experimental data were 0.94 pm, so the accuracy of the measured wavelength of this interrogation system was 0.94 pm, in the dynamic range of 1523.16–1523.2 nm.

## 4. Conclusions

In this paper, we theoretically analyzed the influence of the adjacent output waveguide spacing and length of the FPR on AWG bandwidth. We prepared a high-performance 36-channel AWG. The AWG features a 1.6 nm channel spacing, 3-dB bandwidth of 1.76 nm, −29.76 dB non-adjacent channel crosstalk, and insertion loss range of 3.46 dB. Based on this AWG, we constructed a discrete FBG interrogation system and evaluated its interrogation capabilities. Our findings demonstrate that the system demodulates in the dynamic range of 1522.4–1578.4 nm, covering the C-band. The system achieves a demodulation resolution of 1 pm, with an accuracy of less than 1 pm in the dynamic range of 1523.16–1523.2 nm. This study provides recommendations for improving the PLC-AWG-based FBG wavelength interrogation system. In further studies, the miniaturization of the entire system will be systematically examined.

## Figures and Tables

**Figure 1 nanomaterials-12-02938-f001:**
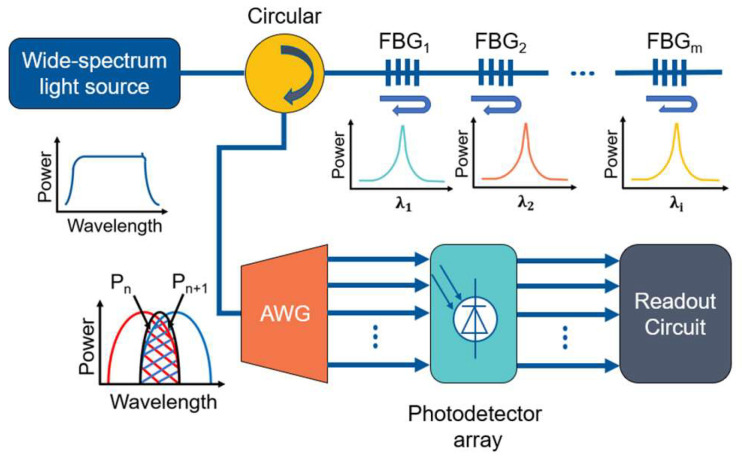
FBG wavelength interrogation system based on AWG.

**Figure 2 nanomaterials-12-02938-f002:**
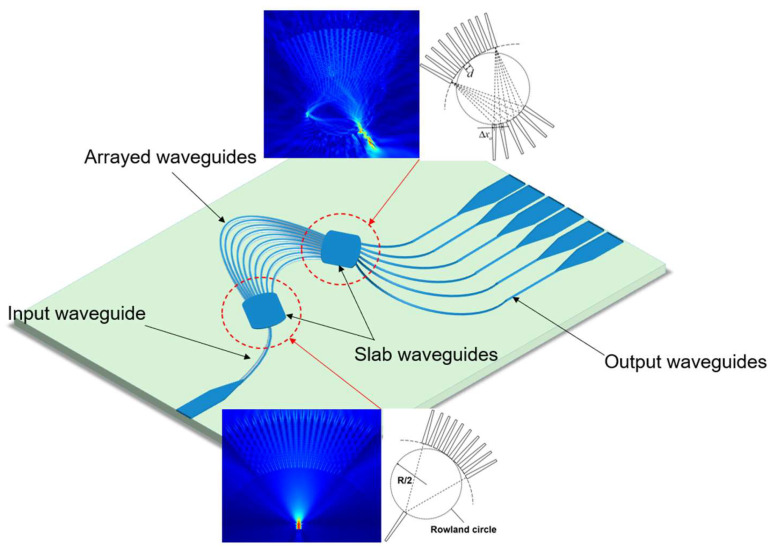
Structural diagram of AWG.

**Figure 3 nanomaterials-12-02938-f003:**
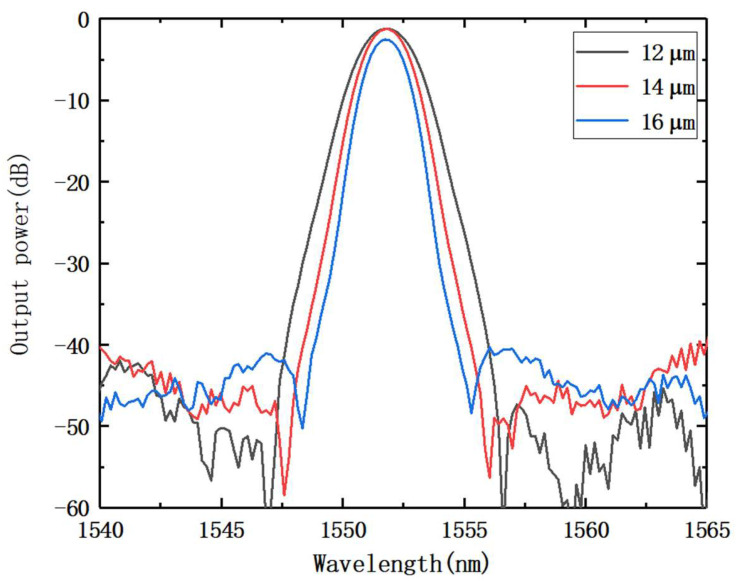
Transmission spectra under different adjacent output waveguide distance.

**Figure 4 nanomaterials-12-02938-f004:**
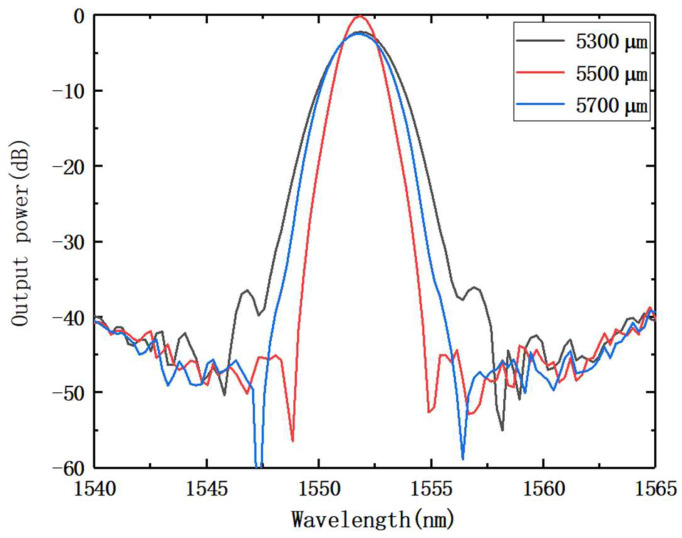
Transmission spectra under different FPR length.

**Figure 5 nanomaterials-12-02938-f005:**
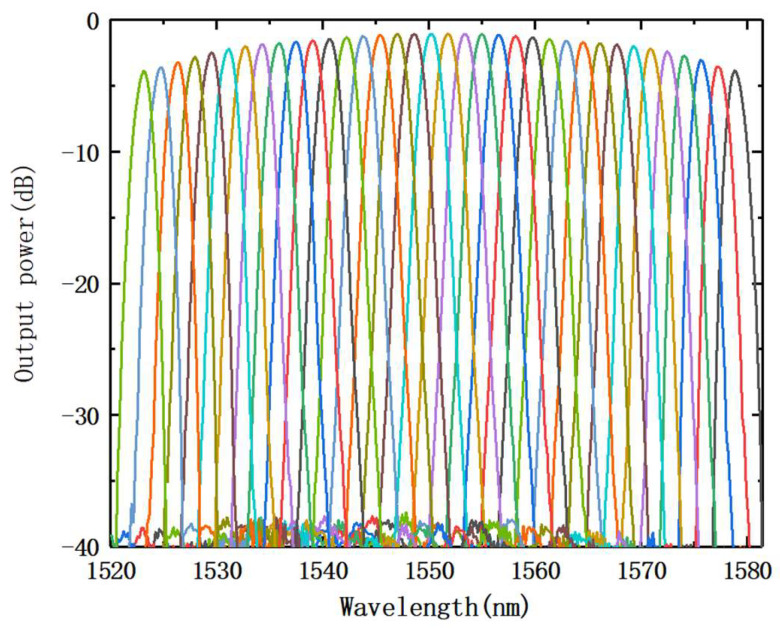
Simulated transmission spectrum of AWG.

**Figure 6 nanomaterials-12-02938-f006:**
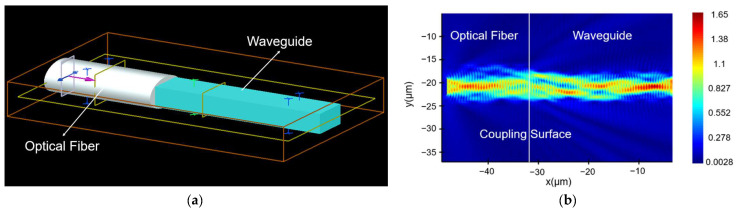
(**a**) Modeling of fiber-waveguide coupling; (**b**) optical field distribution of fiber-waveguide coupling.

**Figure 7 nanomaterials-12-02938-f007:**
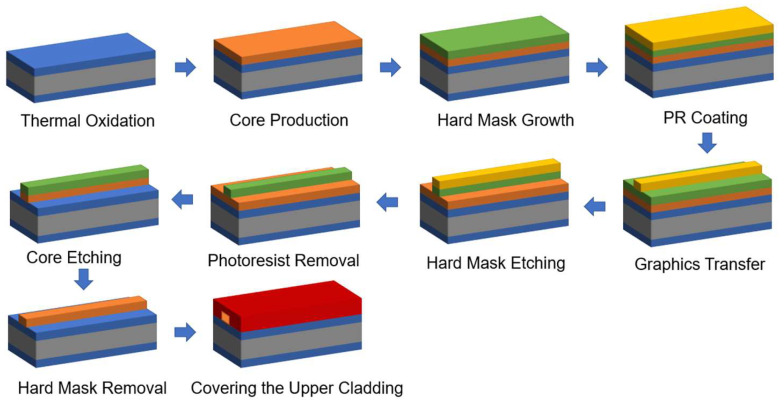
The fabrication process of the AWG.

**Figure 8 nanomaterials-12-02938-f008:**
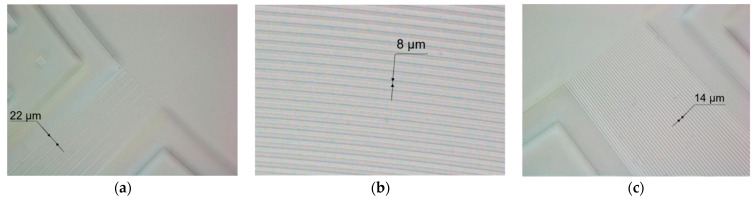
Microscope photo of (**a**) input waveguides; (**b**) arrayed waveguides; (**c**) output waveguides.

**Figure 9 nanomaterials-12-02938-f009:**
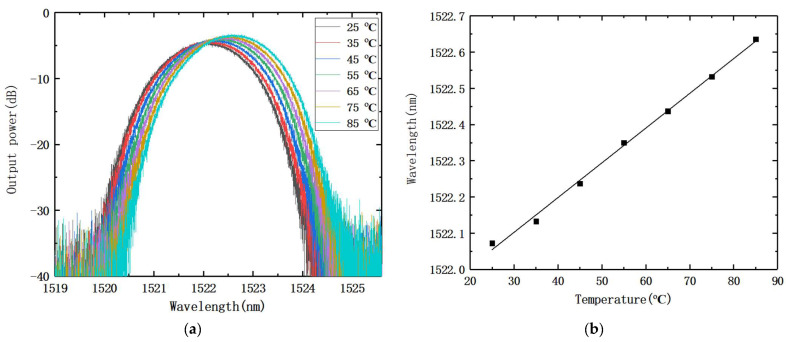
(**a**) Transmission spectra of channel 1 at different temperatures; (**b**) relationship between the center wavelength of channel 1 and temperature.

**Figure 10 nanomaterials-12-02938-f010:**
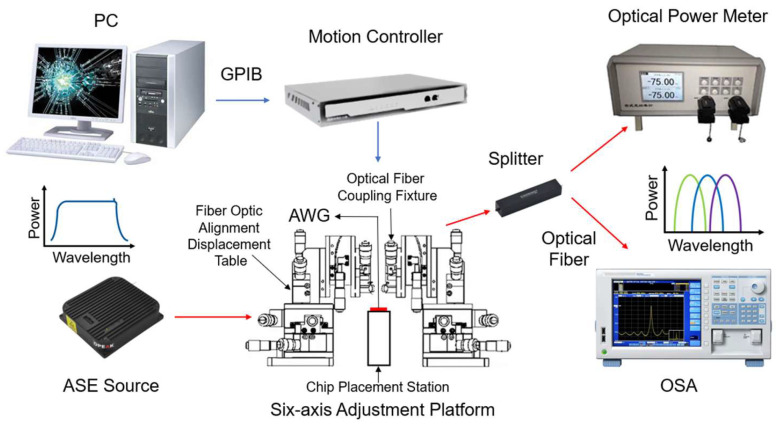
Testing system for the spectrum measurement of the fabricated AWG.

**Figure 11 nanomaterials-12-02938-f011:**
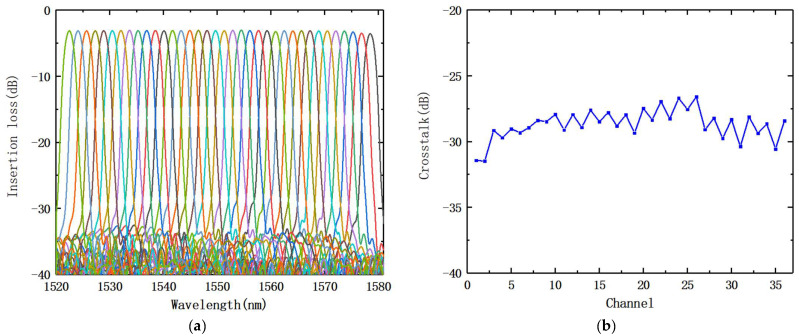
(**a**) Transmission spectra of AWG; (**b**) non-adjacent channel crosstalk of each AWG channel.

**Figure 12 nanomaterials-12-02938-f012:**
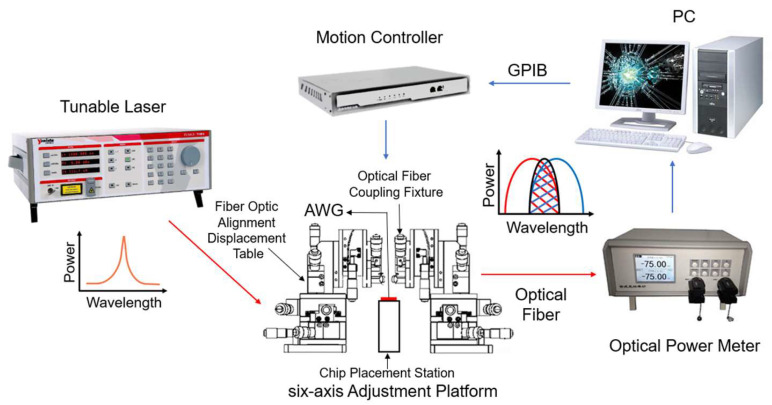
Testing system for measuring the interrogation performance of the fabricated AWG.

**Figure 13 nanomaterials-12-02938-f013:**
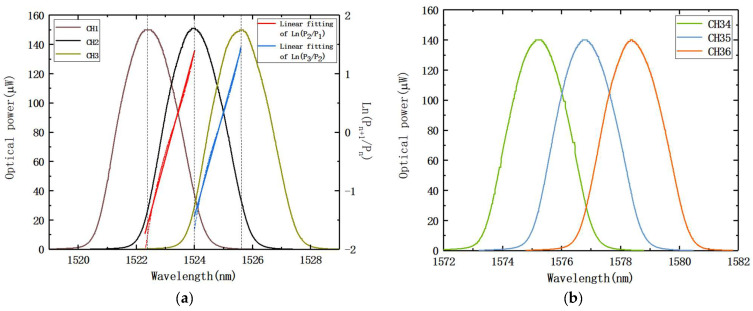
The interrogation results of the experiments employ (**a**) AWG channels 1, 2, and 3; (**b**) AWG channels 34, 35, and 36.

**Figure 14 nanomaterials-12-02938-f014:**
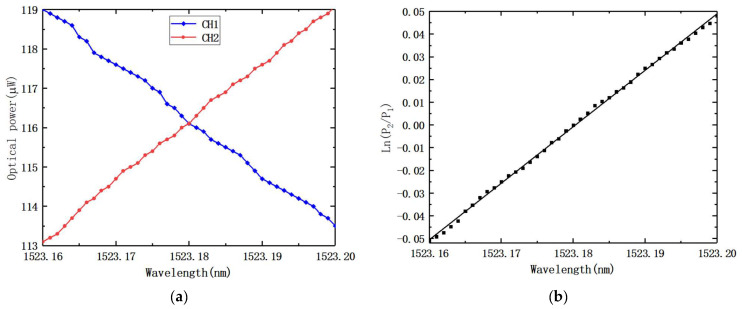
The resolution test of the AWG-based FBG interrogation system: (**a**) the optical power of adjacent AWG channels; (**b**) the logarithm of the output optical power ratio of adjacent AWG channels (channels 1 and 2).

**Table 1 nanomaterials-12-02938-t001:** AWG design key parameters.

Parameters	Value
Central wavelength	1551 nm
Number of input/output ports	7 in/36 out
Adjacent arrayed waveguide spacing	8 µm
Channel spacing	1.6 nm
Adjacent input waveguide distance	22 µm
Adjacent output waveguide distance	14 µm
Rowland circle diameter	5626.669 µm
Adjacent waveguide spacing	19.24235 µm

**Table 2 nanomaterials-12-02938-t002:** Detailed performance measurement of our AWG.

Parameters	Value
3-dB bandwidth	1.76 nm
Insertion loss (IL)	3.46 dB
Crosstalk	−29.76 dB

**Table 3 nanomaterials-12-02938-t003:** The central wavelengths corresponding to AWG channels 1, 2, 3, and 36.

Channel	Center Wavelength
CH1	1522.4 nm
CH2	1524.0 nm
CH3	1525.6 nm
CH36	1578.4 nm

## Data Availability

The data that support the findings of this study are available from the leading author, K.L., upon reasonable request.

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
