# Peer review of "PLC-Based Arrayed Waveguide Grating Design for Fiber Bragg Grating Interrogation System"

_nanomaterials, 2022, doi:10.3390/nano12172938_

Round 1

Reviewer 1 Report

It seems to be that this article is the next one of their already published articles in this and other journals with a similar topic. It means that the research in this area is still growing up by the author's team. From the article's plagiarism point of view, this article seems to be prepared carefully and well. The value of similarity (29%) with other publications is low (please see attachment).
Generally, the article is well prepared and subsequently provides all necessary information about the solved task to a reader. On the other hand, the quality of the figures is low. Therefore, I would recommend changing all of them. I would use figures with better readability and higher resolution. Literature references are sufficient and from quality sources. The English language is at a sufficient level.
The authors should highlight in the paper how their research help in practice.

Remarks to authors:

Please add some details about the temperature stability of your proposed scheme. I think that temperature has a deep impact on your scheme and measured results.

The 1.6 nm channel spacing is unusual, please provide more details on why was used this spacing.

What about crosstalk between each of the outputs of AWG?

Why were used only SiO2 type of materials?

In the Fig. 11b the green curve (CH34) has some " sharper edges" why?

Some details about Coupling efficiency could be there.

Author Response

Dear Reviewer:

Thank you very much for your attention and comments on our paper “PLC-Based Arrayed Waveguide Grating Design for Fiber Bragg Grating Interrogation System”. We have revised the manuscript according to your kind advices and detailed suggestions, and all revised parts have been marked up using the “Track Changes” function in the revised paper.

Our detailed point-by-point responses to the comments are in the attached PDF file. Please see the attachment.

We have revised the manuscript in line with your comments, and we sincerely hope this manuscript will be finally acceptable to be published on Nanomaterials. We appreciate your support very much.

Best regards

Yours sincerely,

Ke Li

Reviewer 2 Report

The work titled "PLC-Based Arrayed Waveguide Grating Design for Fiber Bragg 2 Grating Interrogation System" by Ke Li et al. is a detailed document describing the authors' project of a multifrequency signal demodulation device working in the optical C-band.

Despite the document contains an interesting description of the assembled experimental setup and a dissertation on the retrieved results, it still lacks of some important items and some points need to be fixed to make the draft fully publicable as a scientific article. 

Here it follows a list of notes which I collected during my analysis of the draft, where I mixed suggested minor and major corrections alike:

- Abstract, line 13: please, provide a short definition of an interrogator before introducing it.

- Abstract, line 21: replace "the demodulation of FBG sensor wavelength" with a better statement.

- Ch.1, Introduction, line 33: as above, provide a definition for interrogation.

- Ch.1. Introduction, line 48: "dynamic range of 4000 με": please, check the units of με.

- Ch.1, Introduction, line 55: maybe "AWG, only...." should be written as "AWG: only....".

- Ch.1, Introduction, line 58: I think that "this issue" should be replaced with "these issues", since shortly before there were more than one point to solve.

- Ch.2, Design and Simulation, line 73-75: I believe that the sentece "The nth and (n+1)th channels .... of the nth and (n+1)th channels" can be recast in a more fashionable and less repetitive way.

- Ch.2, Design and Simulation, line 87: "FBG (???G). Where" should be "FBG (???G), where".

- Ch.2, Design and Simulation, line 89-89: the sentence "?n, ?n+1, and ???? are the peak wavelengths of the nth, (n+1)th channels of the AWG" can be rearranged by using a k dummy parameter to represent the generic wavelength and channel as well.

- Ch.2, Design and Simulation, line 98-99: the first "the phase difference," maybe is a typo, and in the case it should be removed.

- Ch.2, Design and Simulation, line 99-101: the sentence "the phase difference of .... guide position, from different output waveguide output" is too repetitive and misleading for the reader; please, recast it in a clearer way.

- Ch.2, Design and Simulation, line 102-103: I suggest the authors to complete fig.2 with another inset showing a geometric scheme of the input and output FPR slab waveguides.

- Ch.2, Design and Simulation: very few details are revealed about the numerical simulation stage; the authors should provide full details of the simulation setup, including the software brand and specific product used, and more importantly some clear details of the numerical setup, like the main method - FDTD, MOM; FEM, etc. -, and the simulation box or the adopted integral equation model. They could start providing details (and a better visual) for the two insets shown in fig.2. 

- Ch.2, Design and Simulation, line 122: please, use a more correct sentence for the caption of fig.3.

- Ch.2, Design and Simulation, line 129-130: maybe the authors made a mistake in the legend; I see strong resemblances between the plots associated to the wavelengths of 5mm and 6mm and a deeper difference between these two and 5.5mm's. Is it right as it is shown in the current manuscript revision, or should it be fixed? And why there are small differences between two of these three plots, while the third one is so different? If everything is coorect, please provide us an explanation; anyway, also add a special mention to this spectral character.

- Ch.2, Design and Simulation, line 130: please, use a more correct sentence for the caption of fig.4.

- Ch.3.2, Measurement, line 162: please, provide a definition for the acronym "ASE".

- Ch.3.2, Measurement, line 182: complete the sentence "in an ultra-clean"

- Ch.3.2, Measurement, line 185: provide a better statement in the place of "is input to".

- Ch.3.2, Measurement, line 178 & 191: I suggest the authors to complete fig.8 and fig,10 with a schematic diagram of the six-axis adjustment platform.    

- Ch.3.2, Measurement, line 192: in the caption of fig.10, use a better replacement for "interrogation performance measurement"

- Ch.4, Conclusions, line 220: please, us e abetter sentence in the place of "the effect law of the adjacent output"; maybe the authors meant "functionality" or "effective performances" for "effective law".

- A final consideration must be spent for the output channels; I realized that each beam component is addressed toward a specific output channel according to its central frequency, with a wavelength spacing of 1.6nm; anyway, the authors provided a single description for the bragg-based output waveguide. Is it correct to assume a single bragg setting for all channles, considering that there is a small frequency gap (i.e., C0*1.6nm / lambda0^2, with lambda0 being the generic central wavelength of each output channel, with C0 being the vacuum light speed) between the 36 channels? Please, add a special paragraph to discuss about this topic and enlight the reader with more details on it.

Author Response

(The authors gave the same response as above.)

Round 2

Reviewer 1 Report

I do not have additional comments on this article. The authors addressed all my comments. The paper is well written and organized for that reason I suggest the article for publishing in the future issue of Nanomaterials MDPI journal.